# Learning Conditioned Graph Structures for Interpretable Visual Question Answering

**Will Norcliffe-Brown**
AimBrain Ltd.
will.norcliffe@aimbrain.com

**Efstathios Vafeias**
AimBrain Ltd.
stathis@aimbrain.com

**Sarah Parisot**
AimBrain Ltd.
sarah@aimbrain.com

## Abstract

Visual Question answering is a challenging problem requiring a combination of concepts from Computer Vision and Natural Language Processing. Most existing approaches use a two streams strategy, computing image and question features that are consequently merged using a variety of techniques. Nonetheless, very few rely on higher level image representations, which can capture semantic and spatial relationships. In this paper, we propose a novel graph-based approach for Visual Question Answering. Our method combines a graph learner module, which learns a question specific graph representation of the input image, with the recent concept of graph convolutions, aiming to learn image representations that capture question specific interactions. We test our approach on the VQA v2 dataset using a simple baseline architecture enhanced by the proposed graph learner module. We obtain promising results with 66.18% accuracy and demonstrate the interpretability of the proposed method. Code can be found at github.com/aimbrain/vqa-project.

## 1 Introduction

Visual Question Answering (VQA) is an emerging topic that has received an increasing amount of attention in recent years [1]. Its attractiveness lies in the fact that it combines two fields that are typically approached individually (Computer Vision and Natural Language Processing (NLP)). This allows researchers to look at both problems from a new perspective. Given an image and a question, the objective of VQA is to answer the question based on the information provided by the image.

Understanding both the question and image, as well as modelling their interactions requires us to combine Computer Vision and NLP techniques. The problem is generally framed in terms of classification, such that the network learns to produce answers from a finite set of classes which facilitates training and evaluation. Most VQA methods follow a two-stream strategy, learning separate image and question embeddings from deep Convolutional Neural Networks (CNNs) and well known word embedding strategies respectively. Techniques to combine the two streams range from element-wise product to bilinear pooling [2, 3] as well as attention based approaches [4].

Recent computer vision works have been exploring higher level representation of images, notably using object detectors and graph-based structures for better semantic and spatial image understanding [5]. Representing images as graphs allows one to explicitly model interactions, so as to seamlessly transfer information between graph items (e.g. objects in the image) through advanced graph processing techniques such as the emerging paradigm of graph CNNs [6, 7, 8, 9]. Such graph based techniques have been the focus of recent VQA works, for abstract image understanding [10] or object counting [11], reaching state of the art performance. Nonetheless, an important drawback of the proposed techniques is the fact that the input graph structures are heavily engineered, image specific rather than question specific, and not easily transferable from abstract scenes to real images. Furthermore, very few approaches provide means to interpret the model's behaviour, an essential aspect that is often lacking in deep learning models.

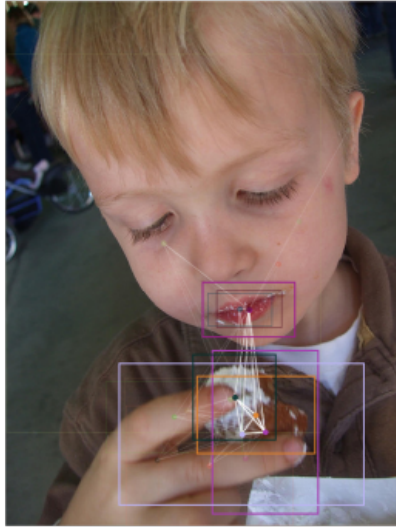 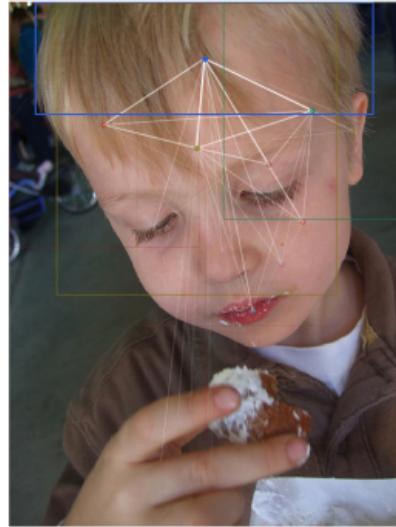

Q: Will the kid leave powdered sugar
on his face?
Prediction: Yes

Q: What color is the kid's hair?
Prediction: Blonde

Figure 1: Epitome of our graph learning module's ability to condition the bounding box connections based on the question. Two graph structures are learned from the same image but tailored to answer different questions. The thickness and opacity of graph nodes (bounding boxes) and edges (white lines) are determined by node degree and edge weights, showing the main objects and relationships of interest.

**Contributions.** In this paper, we propose a novel, interpretable, graph-based approach for visual question answering. Most recent VQA approaches focus on creating new attention architectures of increasing complexity, but fail to model the semantic connections between objects in the scene. Here, we propose to address this issue by introducing a prior in the form of a scene structure, defined as a graph which is learned from observations according to the context of question. Bounding box object detections are defined as graph nodes, while graph edges conditioned on the question are learned via an attention based module. This not only identifies the most relevant objects in the image associated to the question, but the most important interactions (e.g. relative position, similarities) without any handcrafted description of the structure of the graph. Learning a graph structure allows to learn question specific object representations that are influenced by relevant neighbours using graph convolutions. Our intuition is that learning a graph structure not only provides strong predictive power for the VQA task, but also interpretability of the model's behaviour by inspecting the most important graph nodes and edges. Experiments on the VQA v2 dataset confirm our hypothesis. Combined with a relatively simple baseline, our graph learner module achieves 66.18% accuracy on the test set and provides interpretable results via visualisation of the learned graph representations.

## 2 Related work

### 2.1 Graph Convolutional Neural Networks

Graph CNNs (GCNs) are a relatively new concept, aiming to generalise Convolutional Neural Networks (CNNs) to graph structured data. CNNs intrinsically exploit the regular grid-like structure of data defined on the euclidean domain (e.g. images). Extending this concept to non-regularly structured data (e.g. meshes, or social/brain networks) is non trivial. We distinguish graph CNNs defined in the spectral [12, 6] and spatial [8, 9, 13] domains.

Spectral GCNs exploit concepts from graph signal processing [14], using analogies with the Euclidean domain to define a graph Fourier transform, allowing to perform convolutions in the spectral domain as multiplications. Spectral GCNs are the most principled and out of the box approach, however are

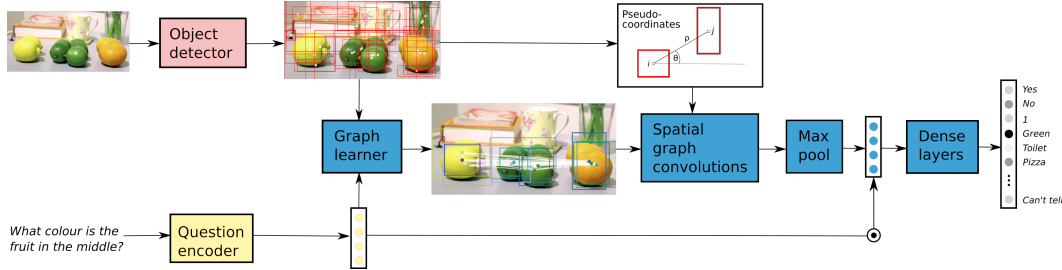

Figure 2: Overview of the proposed model architecture. We model the VQA problem as a classification problem, where each answer from the training set is a class. The core of our method is the graph learner, which takes as input a question encoding, and a set of object bounding boxes with corresponding image features. The graph learner module learns a graph representation of the image that is conditioned on the question, and models the relevant interactions between objects in the scene. We use this graph representation to learn image features that are influenced by their relevant neighbours using graph convolutions, followed by max-pooling, element-wise product and fully connected layers.

limited by the requirement that the graph structure is the same for all training samples, as the trained filters are defined on the graph Laplacian's basis.

Spatial GCNs tend to be more engineered as they require the definition of a node ordering and a patch operator. Several approaches have been defined specifically for regular meshes [13]. Monti et al. [8] recently provided a general spatial GCN formulation, learning a patch operator as a mixture of Gaussians. Recently, Graph Attention Networks were proposed in [9], modelling the convolution operator as an attention operation on node neighbours, where the attention weights can be interpreted as graph edges. Similar to our work, both of these methods compute a type of attention to learn a graph structure. However, while these methods which learn a fixed graph structure, we aim to learn a dynamic graph that is conditioned on the context of a query. Accordingly, our approach extends the notion of an edge to be adaptive to the context.

## 2.2 Visual Question Answering

Explicit modelling of object interactions through graph representations has recently received growing interest. A graph based approach was notably proposed in [10], combining graph representations of questions and abstract images with graph neural networks. This approach beat the state of the art by a large margin, demonstrating the potential of graph based methods for VQA. This approach is however not easily applicable to natural images where the scene graph representation is not known a priori.

The decision to use object proposals as image features resulted in major improvements in VQA performance. This idea was first introduced in [15], outperforming the state of the art with a relatively simple model. Such image representations have since been exploited to model interactions between objects through implicit and explicit graph structures [11, 16], with a focus on counting. [11] compute a graph based on the outer product of the attention weights of proposed features. The computed graph is altered with explicitly engineered features solely to improve their baseline model's ability to count. [16] propose an iterative approach relying on objects similarities to improve the models' counting abilities and interpretability, which was only evaluated on counting questions. The main aim of both approaches is to eliminate duplicate object detections.

## 3   Methods

An overview of the method is shown in Fig. 2. We develop a deep neural network that combines spatial, image and textual features in a novel manner in order to answer a question about an image. Our model first computes a question representation using word embeddings and a Recurrent Neural Network (RNN), and a set of object descriptors comprising bounding box coordinates and image features vectors. Our graph learning module then learns an adjacency matrix of the image objects that is conditioned on a given question. This adjacency matrix enables the next layers - the spatial

graph convolutions - to focus not only on the objects but also on the object relationships that are the most relevant to the question. Our convolved graph features are max-pooled, and combined with the question embedding using a simple element-wise product to predict a class pertaining to the answer of the given question.

## 3.1 Computing model inputs

The first stage of our model is to compute embeddings for both the input image and question. We convert a given image into a set of $K$ visual features using an object detector. Object detections are essential for the subsequent step of our model, as each bounding box will constitute a node in the question specific graph representations we are learning. An embedding is produced for each proposed bounding box, which is the mean of the corresponding area of the convolutional feature map. Using such object features has been observed to yield a better performance in VQA tasks [3, 11], as this allows the model to focus on object-level features rather than a pure CNN which produces a grid of features. For each question, we use pre-trained word embeddings as suggested in [3] to convert the question into a variable length sequence of embeddings. Then we use a dynamic RNN with a GRU cell [17], to encode the sequence of word embeddings as a single question embedding $\mathbf{q}$.

## 3.2 Graph learner

In this section, we introduce the key element of our model and our main contribution: the graph learner module. This novel module produces a graphical representation of an image conditioned on a question. It is general, easy to implement and, as we highlight in Section 4.3, learns complex relationships between features that are interpretable and query dependent. The learned graph structure drives the spatial graph convolutions by defining node neighbourhoods, which in contrast to previous models such as [18, 19], allows unary and pairwise attention to be learned naturally as the adjacency matrix contains self loops.

We seek to construct an undirected graph $\mathcal{G} = \{\mathcal{V}, \mathcal{E}, \mathbf{A}\}$, where $\mathcal{E}$ is the set of graph edges to learn and $\mathbf{A} \in \mathbb{R}^{N \times N}$ the corresponding adjacency matrix. Each vertex $v \in \mathcal{V}$ with $|\mathcal{V}| = N$ corresponds to a detected image object (bounding box coordinates and associated feature vector $\mathbf{v}_n \in \mathbb{R}^d$). We aim to learn the adjacency matrix $\mathbf{A}$ so that each edge $(i, j, A_{i,j}) \in \mathcal{E}$ is conditioned on the question encoding $\mathbf{q}$. Intuitively, we need to model the similarities between feature vectors as well as their relevance to the given question. This is done by first concatenating the question embedding $\mathbf{q}$ onto each of the $N$ visual features $\mathbf{v}_n$, which we write as $[\mathbf{v}_n \| \mathbf{q}]$. We then compute a joint embedding as:

$$\mathbf{e}_n = F([\mathbf{v}_n \| \mathbf{q}]), \qquad n = 1, 2, ..., N \tag{1}$$

where $F : \mathbb{R}^{d_v + d_q} \to \mathbb{R}^{d_e}$ is a non-linear function and $d_v$, $d_q$, $d_e$ are the dimensions of the image feature vectors, question encoding and joint embedding respectively. By concatenating the joint embeddings $\mathbf{e_n}$ together into a matrix $\mathbf{E} \in \mathbb{R}^{N \times d_e}$, it is then possible to define an adjacency matrix for an undirected graph with self loops as $\mathbf{A} = \mathbf{E}\mathbf{E}^T$ so that $A_{i,j} = \mathbf{e}_i^T \mathbf{e}_j$.

Such definition does not impose any constraints on the graph sparsity, and could therefore yield a fully connected adjacency matrix. Not only is this a problem computationally, but the vast majority of VQA questions requires attending to only a small subset of the graph nodes. The learned graph structure will be the backbone of the subsequent graph convolution layers, where the objective is to learn a representation of object features that is conditioned on the most relevant, question-specific neighbours. This requires a sparse graph structure focusing on the most relevant aspects of the image. In order to learn a sparse neighbourhood system for each node, we adopt a ranking strategy as:

$$\mathcal{N}(i) = topm(\mathbf{a}_i) \tag{2}$$

where $topm$ returns the indices of the $m$ largest values of an input vector, and $\mathbf{a}_i$ denotes the $i^{th}$ row of the adjacency matrix. In other words, the neighbourhood system of a given node will correspond to the nodes with which it has the strongest connections.

## 3.3 Spatial graph convolutions

Given a question specific graph structure, we then exploit a method of graph convolutions to learn new object representations that are informed by a neighbourhood system tailored to answer the given

question. The graph vertices $\mathcal{V}$ (i.e. the bounding box and their corresponding features) are notably characterised by their location in the image, making the problem of modelling their interactions inherently spatial. Additionally, a lot of VQA questions require that the model has an awareness of the orientation and relative position of features in an image, an issue that many previous approaches have neglected.

As a result, we opt to use a graph CNN approach inspired by [8], operating directly in the graph domain and heavily relying on spatial relationships. Crucially, their method captures spatial information through the use of a pairwise pseudo-coordinate function $\mathbf{u}(i,j)$ which defines, for each vertex $i$, a coordinate system centred at $i$, with $\mathbf{u}(i,j)$ being the coordinates of vertex $j$ in that system. Our pseudo-coordinate function $\mathbf{u}(i,j)$ returns a polar coordinate vector $(\rho,\theta)$, describing the relative spatial positions of the centres of the bounding boxes associated with vertices $i$ and $j$. We considered both Cartesian and polar coordinates as input to the Gaussian kernels and observed that polar coordinates worked significantly better. We posit that this is because polar coordinates separate orientation ($\theta$) and distance ($\rho$), providing two disentangled factors to represent spatial relationships.

An essential step and challenge of graph CNNs is the definition of a patch operator describing the influence of each neighbouring node that is robust to irregular neighbourhood structures. Monti et al. [8] propose to do so using a set of $K$ Gaussian kernels of learnable means and covariances, where the mean is interpretable as a direction and distance in pseudo coordinates. We obtain a kernel weight $w_k(\mathbf{u})$ for each $k$, such that the patch operator is defined at kernel $k$ for node $i$ as:

$$\mathbf{f}_k(i) = \sum_{j \in \mathcal{N}(i)} w_k(\mathbf{u}(i,j))\mathbf{v}_j, \qquad k = 1, 2, ..., K \tag{3}$$

where $\mathbf{f}_n(i) \in \mathbb{R}^{d_v}$ and $\mathcal{N}(i)$ denotes the neighbourhood of vertex $i$ as described in Eq. 2. Considering a given vertex $i$, we can think of the output of the patch operator as a weighted sum of the neighbouring features, where the set of Gaussian kernels describe the influence of each neighbour on the output of the convolution operation.

We adjust the patch operator so that it includes an additional weighting factor conditioned on the produced graph edges:

$$\mathbf{f}_k(i) = \sum_{j \in \mathcal{N}(i)} w_k(\mathbf{u}(i,j))\mathbf{v}_j \alpha_{ij} \tag{4}$$

with $\alpha_{ij} = s(\mathbf{a}_i)_j$ where $s(.)_j$ is the $j^{th}$ element of a scaling function (defined here as a softmax of the selected adjacency matrix elements). This more general form means that the strength of messages passed between vertices can be weighted by information in addition to spatial orientation. For our use case, this can be interpreted as how much attention the network should pay to the relationship between two nodes in terms of answering a question. Thus the network learns to attend on the visual features in a pairwise manner conditioned on the question.

Finally, we define the output of the convolution operation at vertex $i$ as a concatenation over the K kernels:

$$\mathbf{h}_i = \bigg\|_{k=1}^{K} \mathbf{G}_k \mathbf{f}_k(i) \tag{5}$$

where each $\mathbf{G}_k \in \mathbb{R}^{\frac{d_h}{K} \times d_v}$ is a matrix of learnable weights (the convolution filters), with $d_h$ as the chosen dimensionality of the outputted convolved features. This results in a convolved graph representation $\mathbf{H} \in \mathbb{R}^{N \times d_h}$.

### 3.4 Prediction layers

Our convolved graph representation $\mathbf{H}$ is computed through L spatial graph convolution layers. We then compute a global vector representation of the graph $\mathbf{h}_{max}$ via a max-pooling layer across the node dimension. This operation was chosen so as to get a permutation invariant output, and for its simplicity, so the focus is on the impact of the graph structure. This vector representation of the graph can be considered a highly non-linear compression of the graph, where the representation has been optimised for answering the question at hand. We then merge question $\mathbf{q}$ and image $\mathbf{h}_{max}$ encodings

through a simple element-wise product. Finally, we compute classification logits through a 2-layer MLP with ReLU activations.

## 3.5 Loss function

The VQA task is cast as a multi-class classification problem, where each class corresponds to one of the most common answers in the training set. Following [3], we use a sigmoid activation function with soft target scores, which has been shown to yield better predictions. The intuition behind this is that it allows to consider multiple correct answers per question and provides more information regarding the reliability of each answer. Assuming each question is associated with $n$ valid provided answers, we compute the soft target score of each class as $t = \frac{number\ of\ votes}{n}$. If an answer is not in the top answers (i.e. the considered classes) then it has no corresponding element in the target vector. We then compute the multi-label soft loss which is simply the sum of the binary cross entropy losses for each element in the target vector.:

$$L(\mathbf{t}, \mathbf{y}) = \sum_i t_i \log(1/(1 + exp(-y_i))) + (1 - t_i) \log(exp(-y_i)/(1 + exp(-y_i))) \quad (6)$$

where $y$ is the logit vector (i.e. the output of our model).

# 4 Evaluation

## 4.1 Dataset and preprocessing

We evaluate our model using the VQA 2.0 dataset [20] which contains a total of 1,105,904 questions and about 204,721 images from the COCO dataset. The dataset is split up roughly into proportions of 40%, 20%, 40% for train, validation and test sets respectively. Each question in the dataset is associated with 10 different answers obtained by crowdsourcing. Accuracy on this dataset is computed so as to be robust to inter-human variability as:

$$acc(a) = \min\{\frac{\text{number of times a is chosen}}{3}, 1\} \quad (7)$$

We consider the 3000 most common answers in the train set as possible answers for our network to predict. Each question is tokenized and mapped into a sequence of 300-dimensional pre-trained GloVe word embeddings [21]. The COCO images are encoded as set of 36 object bounding boxes with corresponding 2048-dimensional feature vectors as described in [15]. We normalise the bounding box corners by the image height and width so they lie in the interval $[0, 1]$. The bounding box corners, which provide absolute spatial information, are then concatenated onto the image feature vectors so they become 2052 dimensions ($d_v = 2052$). Pseudo-coordinates are computed as the polar coordinates of the bounding box centres and give the model relative spatial information.

## 4.2 Implementation

Our question encoder is a dynamic Gated Recurrent Unit (GRU) [17] with a hidden state size of 1024 ($d_q = 1024$). Our function $F$ (see Eq. 1), which learns the adjacency matrix, comprises two dense linear layers of size 512 ($d_g = 512$). We use L=2 spatial graph convolution layers of dimensions 2048 and 1024 so that ($d_{h_1} = 2048, d_{h_2} = 1024$). All dense layers and convolutional layers are activated using Rectified Linear Unit (ReLU) activation functions. During training we use dropout on the image features and all but the final dense layers' nodes with a 0.5 probability. We train for 35 epochs using batch size of 64 and the Adam optimizer [22] with a learning rate of 0.0001 which we halve after the 30th epoch. Parameters are chosen based on the performance on the validation set using the training set. The model is then trained on the training and validation sets using the chosen parameters for evaluation on the test set.

## 4.3 Results

We investigated the influence of our model's main parameters on the classification accuracy, namely neighbourhood size ($m$), and the number of Gaussian kernels ($K$). We trained models for $m \in 8 - 32$, with a step of 4; and $K \in \{2, 4, 8, 16, 32\}$. Results are reported in Fig. 3. Our results suggest that

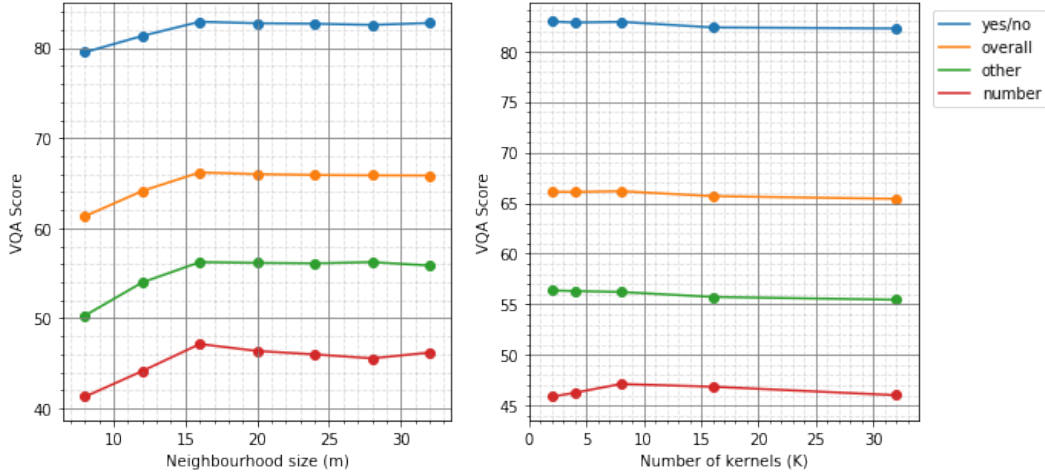

Figure 3: Results of parameter exploration. The VQA score for each question type is reported for a variety of settings of the number of kernels $(K)$ and the neighbourhood size $(m)$. The left hand plot shows varying $m$ while keeping $K = 8$, the right hand plot shows varying $K$ while keeping $m = 16$.

Table 1: VQA 2.0 standard test set results - comparison with baselines and current state of the art methods

| Answer type | All | Y/N | Num. | Other |
|---|---|---|---|---|
| ReasonNet [19] | 64.61 | 78.86 | 41.98 | 57.39 |
| Bottom-Up [3] | 65.67 | 82.20 | 43.90 | 56.26 |
| Counting module [11] | 68.41 | 83.56 | 51.39 | 59.11 |
| kNN graph | 61.00 | 79.35 | 41.63 | 49.70 |
| Attention | 61.90 | 79.87 | 42.48 | 50.95 |
| **Ours** | 66.18 | 82.91 | 47.13 | 56.22 |

$m = 16$ and $K = 8$ are optimal parameters. Performance drops for $m < 16$ and is stable for $m > 16$, while an optimal performance is seen for $K = 8$, in particular for the number questions. In the remaining experiments, we therefore set $K = 8$ and $m = 16$.

Table 4.2 shows our results on the VQA 2.0 test set. We report results on a single version of our model and compare to recent state of the art VQA methods. We compare our model to [19] (*ReasonNet*), the approach by [3] (*Bottom-Up*) who won the 2017 VQA challenge [1], and the recent approach proposed in [11] (*Counting module*) which focuses on optimising counting questions. To highlight the importance of learning the graph structure, we also report results using a k-nearest neighbour graph (based on distances between bounding box centres) (*kNN graph*) as input to the graph convolution layers, and train a baseline model which replaces the graph learning and convolutions with a simple question to image attention (*Attention*). Even though it is simpler than our approach, the *Attention* model provides a good intuition of the advantage of using a graph structure.

Despite not being heavily engineered to optimise performance, our model's performance is close to state of the art. It notably improves substantially on numeric questions (with the exception of [11], which is specifically designed for this purpose). It should also be noted that both *Bottom-Up* and *Counting module* use the object detector with a variable number of objects per image, while we use a fixed number. Our method compares favourably with our baselines (*Attention*, *kNN graph*). This highlights not only the advantage of learning a graph structure, but also of using an attention based approach, as *kNN graph* is the only method without attention.

Figures 4 and 5 show examples of learned graph structures for multiple questions and images. Figures 4 and 5 show the most important nodes and edges of each learned graph (largest node degree/number of connections and edge weights respectively). GraphCNNs learn feature representations of the graph

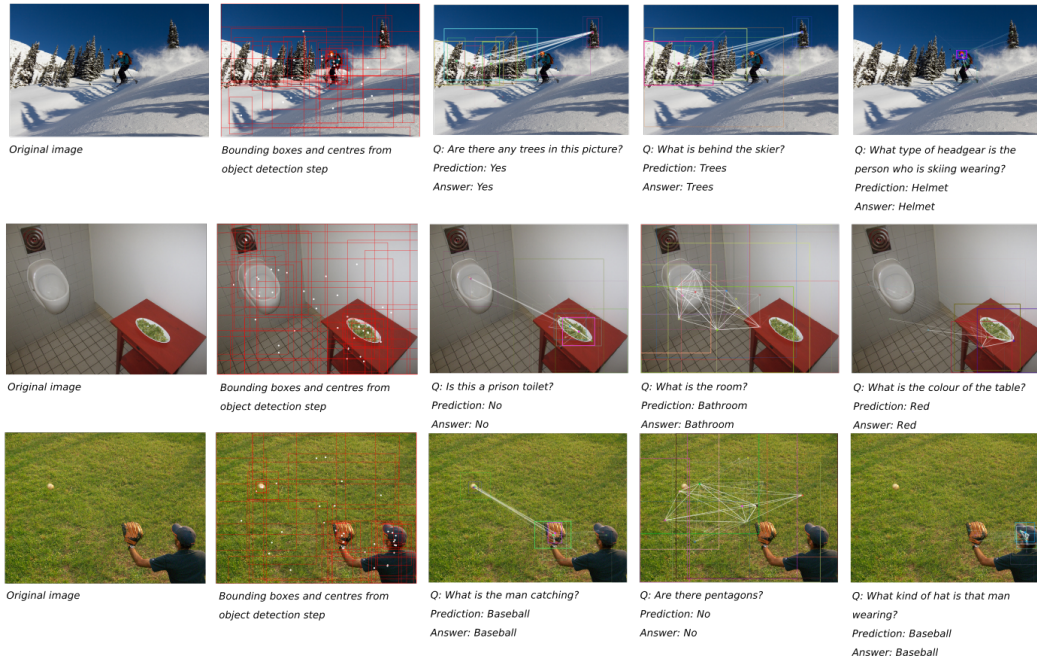

Figure 4: Visual examples of the learned graph structures for multiple images and questions per image. Boxes and edges thickness/opacity correspond to the strengths of the node degree and edge weight respectively, showing the most graph nodes and edges that were considered to be the most relevant to answer the question.

nodes that are influenced by their closest neighbours in the graph. As a result, the most important nodes can be seen as the locations where the network is "looking", as they will strongly influence most feature representations. Edges represent the most important relationships between objects. As a result, one can identify whether the network focussed on the right objects by looking at the most relevant nodes, while edge weights inform of the relationships that were considered as the most relevant to answer the question.

Figure 4 reports examples of graph structures learned leading to successful classification. We report results for multiple questions per image, highlighting how the learned graph is tailored to the question at hand. Figure 5 shows failure cases and allows to study the interpretability of the proposed model. Figures 5-a,d show cases where the model looked at the wrong object, mistakes which can be attributed to missing detected objects (correct purse for Fig. 5-a, man's face for Fig. 5-d). Figure 5-b shows that while the focus is on all animals on the bed, the cat is considered to be the most relevant object, hence the answer. Finally, Fig. 5-c shows a case that may not be adapted to bounding box based models.

## 5    Discussion

In this paper, we propose a novel graph-based approach for Visual Question Answering. Our model learns a graph representation of the input image that is conditioned on the question at hand. It then exploits the learned graph structure to learn better image features that are conditioned on the most relevant neighbours, using the novel and powerful concept of graph convolutions. Experiments on the VQA v2 dataset yield promising results and demonstrate the relevance and interpretability of the learned graph structure.

Several extensions and improvements could be considered. Our main objective was to show the potential and interpretability of learning a graph structure. We found with a fairly simple architecture the learned graph structure was very effective; further work might want to consider more complex architectures to refine the learned graph further. For example, scalar edge weights may not be able to capture the full complexity of the relationships between graph items and so producing vector edges

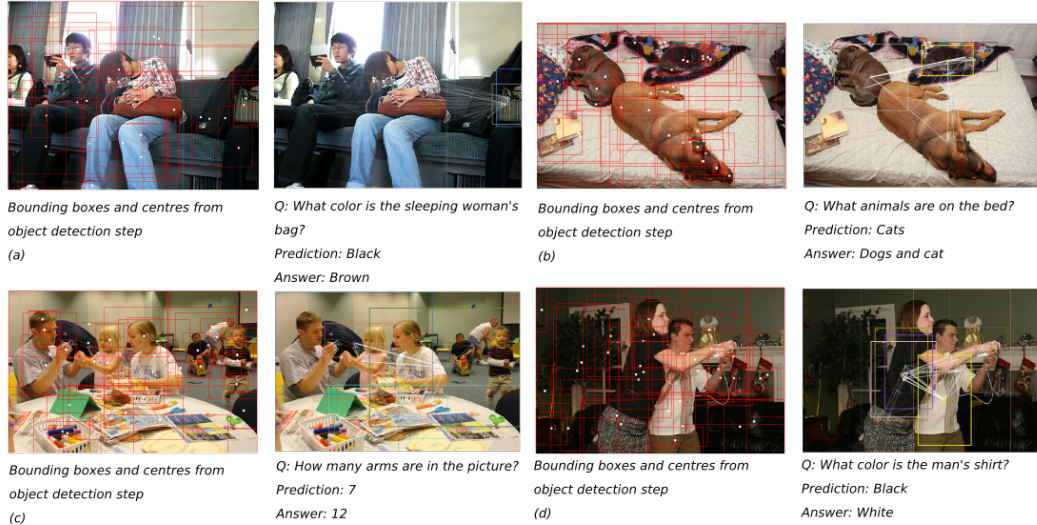

Figure 5: Visual examples of interpretable failures cases. Boxes and edges thickness/opacity correspond to the strengths of the node degree and edge weight respectively, showing the most graph nodes and edges that were considered to be the most relevant to answer the question.

could yield improvements. This could be implemented as an adjacency matrix per convolutional kernel. An important limitation of our approach is the use of an object detector as a preprocessing step. The performance of the model is highly dependent on the quality of the detector, which can yield duplicates or miss objects (as highlighted in the results section). Furthermore, our image features comprise a fixed number of detected objects per image, which can further enhance this problem. Finally, while the focus of this paper is the VQA problem, our approach could be adapted to more general problems, such as few-shots learning tasks where one could learn a graph structure from training samples.

Performance on the VQA v2 dataset is still rather limited, which could be linked to issues within the dataset itself. Indeed, several questions are subjective and cannot be associated with a correct answer (e.g. "Would you want to fly in that plane?") [23]. In addition, modelling the problem as multi-class classification is the most common approach in recent VQA methods, and can strongly limit performance. Questions often require answers that cannot be found in the predefined answers (e.g. "what time is it?"). This explains the low performance of "Number" questions, as it comprises several questions requiring answers absent from the training set.

## Footnotes

[1] http://www.visualqa.org/

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
