[Supplementary Material]

# Learning Conditioned Graph Structures for Interpretable Visual Question Answering Supplementary Material

**Will Norcliffe-Brown**
AimBrain Ltd.
will.norcliffe@aimbrain.com

**Efstathios Vafeias**
AimBrain Ltd.
stathis@aimbrain.com

**Sarah Parisot**
AimBrain Ltd.
sarah@aimbrain.com

## 1 Supplementary Material

We provide additional visual examples of learned graph structures for successful question answering (Figure 1) and interpretable failure cases (Figure 2). Figure 1 shows cases where relational information is required and where the graph structure allows these comparisons. For example, the top left image compares all instances of sheep in the image to verify whether they all are the same colour. Similarly the bottom right image compares the two women to identify which one has the black top and which one is in the back.

Figure 2 highlights multiple failure cases. The first image (top left) shows how the classification setting, while helpful to facilitate training and evaluation, can limit the performance of the model. While the model finds the correct answer (one of the sheep is black), the correct explanation is not available amongst the possible classification outputs.

The remaining examples show how the learned graph structure allows to interpret incorrect answers. Most of these cases are linked to failures from the object detectors, showing that our proposed method is sensitive to missing detections. In the second image (top right) the object detector fails to create a good bounding box around the mouse, possibly because it is out of focus in the background, and therefore only picks up keyboard keys.

The middle left image shows that the model identifies all front facing persons on the image, but misses the person mostly hidden in the background. Once again, this can be linked to the fact that the person in the background is not detected.

In the middle right image, the model identifies the person in the background as wearing a sweater as can shown by the highlighted bounding box.

For the left bottom image, the model correctly counts all of the people visible on the image, but fails to acknowledge that none of them are on a bridge.

In the right bottom image, we can easily see which player is missing from the model's count based on the selected bounding boxes. Contrarily to the central person, the three men in the background are well detected, so the model focuses on these objects. While the central man was partly detected, this suggests that the model needs similar bounding box detections around each of the relevant objects to count accurately.

Q: Are all these animals the same colour?

Prediction: Yes

Answer: Yes

Q: What is this horse pulling?

Prediction: Carriage

Answer: Cart

Q: How many giraffes are drinking?

Prediction: 1

Answer: 1

Q: Are these men twins?

Prediction: No

Answer: No

Q: Is this healthy?

Prediction: Yes

Answer: Yes

Q: Is the lady at the back in black top?

Prediction: No

Answer: No

Figure 1: Visual examples of graph structures leading to successful question answering

Q: What is different about the two sheep's faces?

Prediction: Black

Answer: One is black

Q: What type of device is the black item?

Prediction: Keyboard

Answer: Mouse

Q: How many people are in the picture?

Prediction: 4

Answer: 5

Q: Is anyone wearing a sweater?

Prediction: Yes

Answer: No

Q: How many people are walking across the bridge?

Prediction: 3

Answer: 0

Q: How many players are on the field?

Prediction: 3

Answer: 4

Figure 2: Visual examples of graph structures leading to unsuccessful question answering