[Reviews · NeurIPS 2018]

Reviewer 1



— This paper proposes a graph-based network for visual question answering. — The network first predicts a question-conditioned and image-grounded graph which is then processed by a graph convolutional network to reason about spatial and semantic interactions, followed by a multi-layer perceptron to predict the answer. Strengths — Interesting and novel approach. — Qualitative results look great! The predicted graph connectivity (at least in these few examples) looks quite intuitive and interpretable, even when the model predicts the incorrect answer. Weaknesses — Figure 2 caption says “[insert quick recap here]” :) — The paper emphasizes multiple times that the proposed approach achieves state of the art accuracies on VQA v2, but that does not seem to be the case. The best published result so far — the counting module by Zhang et al., ICLR 2018 — performs ~3% better than the proposed approach (as shown in Table 1 as well). This claim needs to be sufficiently toned down. Also, the proposed approach is marginally better than the base Bottom-Up architecture. Is an improvement of 0.1% on VQA v2 test-std statistically significant? — “Performance on the VQA v2 dataset is still rather limited, which could be linked to issues within the dataset itself.” — this is quite a strong claim, and does not seem to be sufficiently backed with evidence. The 2 reasons given are 1) “several questions require subjective answers”. Agreed, seems plausible, but it would be useful to highlight what % of the error rate is made up of subjective questions, by looking at agreement among humans for example. If that % is significantly high, that could be problematic. 2) “require answers that cannot be found in the current multi-class classification set-up” — this is not an issue with the dataset at all, but the consequence of a particular modeling choice. There is a rich, heavy tail of answers in the real world, much beyond 3k answers most models end up going with. Evaluation Jointly predicting a scene graph and using a graph CNN for reasoning for VQA makes a lot of sense. Results don't outperform state-of-the-art and claims related to that should be toned down appropriately, but the approach is sufficiently promising to deserve publication nonetheless.

Reviewer 2



[Summary] T_x0008_his paper presented a novel graph-based approach for visual question answering. The proposed method first uses object feature and question feature as graph node representation. Then, a spatial graph convolution network is used to learn the image representations that capture the question specific interactions. One nice property is the interpretability of the proposed model by visualizing the learned graph weight. Experiments on VQA v2 dataset demonstrate the effectiveness of the proposed methods. [Strength] 1: The proposed method is simple and effective, with simple model architecture, the proposed method can achieve state of the art performance on VQA dataset. 2: The graph visualization and interpretability of the proposed method is a plus. [Weakness] 1: I like the paper's idea and result. However, this paper really REQUIRE the ablation study to justify the effectiveness of different compositions. For example: - In eq2, what is the number of m, and how m affect the results? - In eq3, what is the dimension of w_n? what if the use the Euclidean coordinate instead of Polar coordinate? - In eq3, how the number of Gaussian kernels changes the experiment results. 2: Based on the paper's description, I think it will be hard to replicate the result. It would be great if the authors can release the code after the acceptance of the paper. 3: There are several typos in the paper, need better proof reading.

Reviewer 3



-------------- Summary: -------------- The submission proposed a graph-based, question-image fusion mechanism for visual question answering. This approach learns an embedding of question and object detection features and uses it to predict question-image dependent adjacency matrices -- essentially a graph structure with nodes corresponding to object detections and edges weighted according to this learned metric. This learned module makes up the core technical contribution of the work as both spatial GCNs and the use of object detection features come from existing work. -------------- Clarity: -------------- [C1] The portions of the method discussed in the submission are relatively clearly described. However, the model description stop abruptly after computing the GCN node representations. relying instead on Figure 2 and L92/L201 to convey the rest of the model (max-pool followed by full-connected layers). While these remaining components are quite simple, the choice of performing a max-pool is interesting but remains unjustified in the current approach. [C2] The baselines are described incredibly quickly and without sufficient detail for the 'attentional' model to understand the comparison made in Table 1. [C3] Significant space in the submission is used to display multiple large example images, Fig 1, 3, and 4, but there is limited discussion of these figures. Further, I'm not sure what are the take-aways from the examples in Fig 3 and 4. The authors point to these images as examples of interpretability, but the graph structure (the novel addition over standard attentional frameworks) doesn't seem to clearly be very interpretable. [C4] Many of the figure captions are uninformative or incomplete. [C5] The paper does not achieve state of the art results as claimed. I don't particularly find small improvements in SoTA to be that good of a measure of a paper anyway, but given that the claim appears in multiple places in the submission, it is worth pointing out. -------------- Quality: -------------- [Q1] I'm all for the diversity of ideas and feel the fact that this submission performs similarly (or somewhat worse than) existing work while introducing a significantly different attentional mechanism is valuable. That said, given the lack of analysis or discussion it is difficult to find insights about the problem from this work. [Q2] The proposed graph learning approach is not well explored. It is unclear what sort of spatial relationships the Guassian GCN kernels are learning. It is unclear what question-conditioned object relationships the graph learner module is representing. It is not discussed what effect the top-k or max-pool design decisions have on the model. It would be good if authors could provide some discussion of these. [Q3] I do not see sufficient discussion or experiments of interpretability to substantiate the claims that the graph based attention model is more interpretable. It would be good for the authors to define what they mean by interpretable and provide some analysis based on human studies. -------------- Originality: -------------- [O1] Some related work on attentional GCNs (i.e GCNs which learn edge weights) is missed. For example, GRAPH ATTENTION NETWORKS from ICLR 2018. I admit that ICLR and this submission deadline are fairy close (~2 weeks) but this work has been publicly available for some time. [O2] The core novelty of this work is in the prediction of question-conditioned edge weights between object nodes in the graph structure. To my knowledge, this is fairly novel in the context of VQA. -------------- Significance: -------------- [S1] Without additional analysis to identify the behavior, strengths, and weaknesses of the proposed method, I don't think the current submission would significantly advance understanding on the topic of VQA or question-conditioned attention mechanisms. Which is unfortunate given the direction itself may be valuable. Post Rebuttal: The author response addressed some of my concerns and I've updated my recommendation accordingly. I urge the authors to put concerted effort into explore and exposing all the interesting design decisions and model behaviors of their new architecture in the final draft. The proposed solution is a fairly significant shift from existing approach that performs fairly similarly and it is useful to the community to illuminate why.